# Mix, Don't Pick: Why Synthetic Corpus Composition Matters for Time Series Foundation Model Pretraining

**Aaryan Nagpal** [1] **Debdeep Sanyal** [1] **Murari Mandal** [1 2] **Dhruv Kumar** [1 3 †] **Saurabh Deshpande** [1 †]

## Abstract

Choosing the wrong synthetic generator for time-series foundation model pretraining is costly: under identical training budgets, the best and worst generators produce up to a $2\times$ gap in forecasting error, yet the field has no principled way to make this choice. The problem is compounded by the fact that generator rankings are not stable across architectures: across 11 generator families evaluated on Chronos-T5-Mini and Moirai-Small trained from scratch, we find that which generators are useful depends on the model architecture. Rather than solving the generator selection problem, we sidestep it: a simple equal-weight mixture of all generators matches or beats the best individual generator for both architectures, and composing this mixture with real data yields the strongest pretraining corpora overall. Synthetic pretraining is therefore a corpus composition problem, not a generator selection problem, and composition choices should be validated per model family rather than assumed to transfer.

## 1. Introduction

Time-series foundation models (TSFMs) are increasingly pretrained on synthetic data, either as the primary source (Moroshan et al., 2025) or as augmentation alongside real corpora (Ansari et al., 2024), motivated by the scarcity and licensing constraints of real-world data. Yet the choice of which generators to use and how to combine them is typically made ad hoc. Generator selection is driven by convention or convenience rather than systematic evaluation, and the broader question of how to compose a pretraining corpus, including which generators to include, whether to mix families, and how much real data to blend in, remains largely unexamined.

A natural approach is to select generators whose output most closely resembles real time series, an assumption implicit in feature-based validation work that measures whether synthetic series match real statistical structure (Bahrpeyma et al., 2021; Fulcher & Jones, 2017; Lubba et al., 2019). However, resemblance to real data and usefulness for pretraining are distinct properties.

Existing work on synthetic TSFM pretraining (Ansari et al., 2024; Das et al., 2024; Moroshan et al., 2025; Woo et al., 2024) has not systematically tested whether fidelity-based generator selection translates to downstream performance, whether generator rankings transfer across model architectures, or how synthetic and real data should be balanced in a pretraining corpus.

We argue that these are not isolated questions but facets of a single corpus composition problem, analogous to the data mixing challenges studied in large language model pretraining, where corpus design decisions have been shown to substantially affect downstream quality (Xie et al., 2023).

To test this, we conduct a systematic evaluation of 11 synthetic generator families spanning statistical, stochastic, dynamical, and waveform-based processes. We train Chronos-T5-Mini (Ansari et al., 2024) and Moirai-Small (Woo et al., 2024) from scratch on each generator individually and on an equal-weight mixture of all 11 generators. We then study real–synthetic composition by blending a real reference corpus with the synthetic mixture at varying ratios. The real reference is sampled from the GIFT-Eval pretraining pool (Aksu et al., 2024), with train–test leakage avoided by construction. All models are evaluated zero-shot on the 28-dataset GIFT-Eval benchmark.

Our study yields three main findings:

1. Synthetic generators are not interchangeable: across 11 generator families, downstream utility varies widely under identical training budgets, and generator rankings differ substantially between Chronos-T5-Mini and Moirai-Small.

Code available at https://github.com/birla-ai-labs/mix-dont-pick. † Equal supervision. [1]Birla AI Labs, Mumbai, India [2]KIIT, Bhubaneswar, India [3]BITS Pilani, Pilani, India. Correspondence to: Saurabh Deshpande <saurabh.deshpande-c@oab.adityabirla.com>, Dhruv Kumar <dhruv.kumar-c@oab.adityabirla.com>.

*ICML 2026 Workshop on Foundation Models for Structured Data*, Seoul, South Korea. 2026. Copyright 2026 by the author(s).

2. An equal-weight mixture of all generators provides a robust synthetic-only default: it matches the strongest individual generator for Moirai-Small and improves over it for Chronos-T5-Mini.

3. Composing real and synthetic data can match or improve over pure-source baselines for both architectures, with the strongest corpora combining the two. The optimal ratio is architecture-dependent, motivating per-model validation of corpus composition.

## 2. Experimental Design

We evaluate synthetic time-series pretraining as a corpus composition problem. Our design trains two TSFM architectures from scratch on each of 11 synthetic generators individually, on multi-generator and real–synthetic mixtures, and on a real reference corpus, then evaluates all models zero-shot on a shared benchmark.

### 2.1. Synthetic and Real Corpora

We construct synthetic corpora from 11 time-series generators (see Appendix C), producing 11.2B time points in total. Each generator-specific corpus contains 1M univariate windows of length 1024. The generator pool spans linear statistical models such as ARIMA and ETS (Box & Jenkins, 1970; Hyndman et al., 2008), stochastic process families including fBm, SDEs, and GARCH-type volatility models (Mandelbrot & Van Ness, 1968; Uhlenbeck & Ornstein, 1930; Gillespie, 1996; Bollerslev, 1986), deterministic chaotic systems (Lorenz, 1963; Mackey & Glass, 1977), composite signal generation through TimeSynth (Maat et al., 2017), trend-seasonality-irregularity decomposition (Bahrpeyma et al., 2021), structural and non-smooth periodic generators inspired by StepFunction and Waveform families (Moroshan et al., 2025), and Gaussian-process based KernelSynth generators (Ansari et al., 2024; Rasmussen & Williams, 2006). Full generator descriptions and hyperparameter distributions are provided in Appendix A.

For real data, we construct a reference corpus from the GIFT-Eval pretraining dataset (Aksu et al., 2024), which is aligned with the downstream GIFT-Eval benchmark while avoiding train–test leakage. Because available long-context real series are unevenly distributed across domains, this reference corpus is availability-constrained rather than uniformly balanced. We sample 1M univariate windows of length 1024, stratifying by frequency and domain where possible. To reduce dominance by large sources, we impose minimum allocations for viable strata and per-dataset caps. We exclude series shorter than 1024. Full sampling details are provided in Appendix D.

We train on all 11 single-generator corpora, an equal-weight mixture of all 11 generators (Mixed11), the real reference corpus, and real–synthetic mixtures combining the real reference with Mixed11 at 75-25, 50-50, and 25-75 real-synthetic ratios by window count. All training corpora are matched at 1M windows of length 1024. Appendix C provides the full corpus manifest.

### 2.2. Models and Evaluation

We train Chronos-T5-Mini (Ansari et al., 2024) and Moirai-Small (Woo et al., 2024) from scratch on each corpus under matched training budgets. Full training configuration is in Appendix E.

We evaluate all models zero-shot on the 28-dataset GIFT-Eval benchmark. Because many datasets are evaluated at multiple forecast horizons, paired bootstrap comparisons use dataset–horizon tasks as the resampling unit, yielding 97 task-level observations. We report normalised CRPS (Gneiting & Raftery, 2007), a probabilistic forecasting error, and normalised MASE (Hyndman & Koehler, 2006), a scale-free point-forecasting error. Both are normalised against the seasonal-naive baseline, so lower is better and values below 1 indicate improvement over that baseline.

## 3. Results

We organise the results around three questions. First, how much does the choice of synthetic generator matter under a fixed training budget? Second, can a simple multi-generator mixture provide a robust alternative to selecting a single generator? Third, when real data is available, does adding synthetic data help, or does it merely dilute the real corpus? Across both architectures, the answer is that corpus composition has a first-order effect on zero-shot forecasting performance.

### 3.1. Single-Generator Corpora

A natural first strategy is to identify a strong synthetic generator and pretrain on it. To evaluate this strategy, we train each architecture on each synthetic generator separately under the same corpus size and training budget. Table 1 reports the resulting zero-shot performance, alongside the real reference corpus as an anchor.

Under identical training budgets, generator choice has a substantial effect: CRPS varies by a factor of 1.6 across generators for Moirai-Small and 2.1 for Chronos-T5-Mini. The ranking also does not transfer cleanly across architectures or metrics. KernelSynth is the strongest generator by CRPS for both architectures, but SDE gives the best synthetic MASE for Chronos-T5-Mini. More strikingly, ETS is the second-best generator for Moirai-Small by CRPS, yet it is the weakest generator for Chronos-T5-Mini. These results suggest that single-generator selection is a model-dependent decision rather than a universally transferable recipe.

*Table 1*. Zero-shot GIFT-Eval performance. Scores are geometric means across 97 evaluation tasks. Lower is better; values below 1.000 beat seasonal naïve. **Bold** marks the best single-generator corpus for each model and metric. Underline: second-best per model (ETS for Moirai-Small; SDE for Chronos-T5-Mini). Sorted by Moirai-Small CRPS.

| | Moirai-Small | | Chronos-T5-Mini | |
| --- | --- | --- | --- | --- |
| Generator | CRPS | MASE | CRPS | MASE |
| **KernelSynth** | **0.734** | **1.049** | **0.936** | 1.290 |
| ETS | 0.820 | 1.154 | 1.976 | 2.694 |
| Waveform | 0.870 | 1.240 | 1.328 | 1.841 |
| SDE | 0.910 | 1.306 | 0.981 | **1.282** |
| StepFunction | 0.958 | 1.391 | 1.048 | 1.359 |
| TSI | 0.969 | 1.369 | 1.195 | 1.679 |
| ARIMA | 0.988 | 1.430 | 1.128 | 1.916 |
| fBm | 1.036 | 1.419 | 1.681 | 2.495 |
| GARCH | 1.055 | 1.466 | 1.108 | 1.863 |
| Chaotic | 1.154 | 1.636 | 1.842 | 2.867 |
| TimeSynth | 1.194 | 1.659 | 1.153 | 1.884 |
| Real Reference | 0.814 | 1.149 | 0.791 | 1.061 |

## 3.2. Multi-Generator Mixing

The single-generator sweep shows that generator rankings can depend on the target architecture. This raises a practical question: instead of selecting one generator, can we obtain a more robust synthetic pretraining corpus by mixing diverse generators? We test this with Mixed11, an equal-weight mixture over all 11 generator families. Table 2 compares Mixed11 with the strongest single-generator baselines identified in the previous section.

*Table 2*. Mixed11 compared with the strongest single-generator baselines for each architecture. Scores are geometric means across 97 evaluation tasks. Lower is better.

| Model | Condition | CRPS | MASE |
| --- | --- | --- | --- |
| Moirai-Small | KernelSynth | **0.734** | **1.049** |
| | ETS | 0.820 | 1.154 |
| | Mixed11 | 0.735 | 1.069 |
| Chronos-T5-Mini | KernelSynth | 0.936 | 1.290 |
| | SDE | 0.981 | 1.282 |
| | Mixed11 | **0.906** | **1.171** |

For Moirai-Small, Mixed11 is essentially tied with KernelSynth on CRPS and remains clearly ahead of the next-best generator, ETS. For Chronos-T5-Mini, the benefit is stronger: Mixed11 improves over both KernelSynth and SDE on CRPS, and also gives the best MASE among the displayed synthetic corpora. Thus, mixing does not simply average away useful structure from strong generators. Instead, it provides a robust synthetic-only corpus that is competitive with, and in some cases better than, the best individual generator observed in the sweep.

To verify that this comparison is not driven only by small aggregate differences, we isolate the paired bootstrap comparison between KernelSynth and Mixed11. This is the most direct comparison for the synthetic-only setting because KernelSynth is the strongest single-generator baseline by CRPS in the seed-42 sweep, while Mixed11 tests whether combining generators can preserve or improve over the best individual source. Full single-generator and composition bootstrap results are reported in Appendix G.

*Table 3*. Seed-42 paired bootstrap comparison between KernelSynth and Mixed11. Relative deltas are reported for KernelSynth versus Mixed11; negative values indicate that KernelSynth has lower error, while positive values indicate that Mixed11 has lower error.

| Model | Metric | Statistic | Value |
| --- | --- | --- | --- |
| Moirai-Small | CRPS | Rel. Δ | -0.2% |
| | | 95% CI | [-3.7%, 3.3%] |
| | | Win rate | 0.48 |
| | MASE | Rel. Δ | -1.9% |
| | | 95% CI | [-5.5%, 1.4%] |
| | | Win rate | 0.47 |
| Chronos-T5-Mini | CRPS | Rel. Δ | 3.3% |
| | | 95% CI | [-1.2%, 8.2%] |
| | | Win rate | 0.42 |
| | MASE | Rel. Δ | 10.2% |
| | | 95% CI | [5.5%, 15.2%] |
| | | Win rate | 0.34 |

For Moirai-Small, KernelSynth and Mixed11 are effectively tied: both CRPS and MASE confidence intervals overlap zero. For Chronos-T5-Mini, Mixed11 is stronger, especially on MASE, where the confidence interval is entirely above zero in favour of Mixed11. This supports the broader conclusion that multi-generator mixing can preserve the benefits of a strong individual generator while improving robustness across model families.

## 3.3. Real–Synthetic Composition

Mixed11 provides a strong synthetic-only default, but in practical pretraining settings synthetic data is rarely considered in isolation. The more relevant question is whether synthetic data can complement an available real corpus, or whether it simply dilutes useful real-world structure. We therefore blend the real reference corpus with Mixed11 at three window ratios. For each mixture, we train three independent model initialisations on the same fixed corpus, so the reported variation reflects model-level stochasticity rather than corpus resampling. Table 4 reports the resulting zero-shot performance.

For Moirai-Small, adding synthetic data to the real reference corpus gives a clear gain: the 75–25 real–synthetic mixture

*Table 4.* Real–synthetic mixture performance on GIFT-Eval. Scores are geometric means across evaluation datasets, reported as mean ± std over three model initialisations. REAL REF and MIXED11 are the pure-source endpoints; intermediate rows give the real–synthetic window ratio. Lower is better; values below 1.000 beat seasonal naïve. **Bold** marks the lowest mean CRPS per model.

| Model | R-S ratio | CRPS ↓ | MASE ↓ |
|---|---|---|---|
| Moirai-Small | Real Ref | 0.830±0.014 | 1.172±0.021 |
| | 75-25 | **0.685±0.015** | **0.984±0.023** |
| | 50-50 | 0.710±0.015 | 1.012±0.023 |
| | 25-75 | 0.775±0.012 | 1.100±0.021 |
| | Mixed11 | 0.733±0.014 | 1.058±0.022 |
| Chronos-T5-Mini | Real Ref | 0.779±0.010 | 1.052±0.008 |
| | 75-25 | 0.794±0.011 | 1.052±0.009 |
| | 50-50 | **0.772±0.016** | 1.019±0.014 |
| | 25-75 | 0.780±0.010 | 1.044±0.012 |
| | Mixed11 | 0.899±0.015 | 1.165±0.006 |

improves over both pure endpoints on CRPS and MASE. This suggests that the synthetic mixture is not merely substituting for missing real data, but adding complementary structure under the fixed training budget. For Chronos-T5-Mini, the picture is more conservative. The 50–50 mixture gives the lowest mean CRPS and MASE, but the CRPS bootstrap interval against the real reference overlaps zero; we therefore interpret this as parity on CRPS and a modest gain on MASE rather than a decisive win. Overall, the best real–synthetic ratio is architecture-dependent, and the domain-level results in Appendix H show that this dependence also varies across domains and horizons.

## 4. Conclusion

We studied synthetic time-series pretraining as a corpus composition problem. Across 11 generator families and two architectures, generator choice produces a factor-of-two spread in forecasting error under identical training budgets, and generator rankings differ across architectures and metrics. An equal-weight mixture of all generators sidesteps much of this selection problem, matching the strongest individual generator for Moirai-Small and improving over it for Chronos-T5-Mini. Real–synthetic mixtures clearly improve over both pure-source baselines for Moirai-Small and reach parity or modest improvement over the real reference for Chronos-T5-Mini.

These results support a practical lesson: pretraining corpora should be designed and reported explicitly, including generator choice, mixture design, and real–synthetic ratio. Corpus composition is a first-order design decision, not a detail to be fixed by convention. In particular, synthetic data should not be treated as a monolithic category. Different generators encode different temporal structures, and their usefulness depends on how those structures interact with the model architecture and the downstream evaluation distribution.

**Limitations:** Our study is limited to two compact TSFM architectures, a fixed 1M-window training budget, and a small number of global real–synthetic mixture ratios. Larger models, longer training, or different downstream benchmarks may change the optimal corpus composition. Moreover, the domain-level results suggest that the best composition may vary by domain and horizon rather than following a single global ratio.

**Future work:** A natural next step is to estimate the marginal contribution of each generator through leave-one-generator-out or learned mixture-weight experiments. More adaptive corpus construction could also condition mixture weights on domain, frequency, horizon, or model family rather than using a single global ratio. Finally, future work should connect feature-space diagnostics of synthetic generators with downstream utility, so that corpus design can move from exhaustive empirical sweeps toward predictive rules for selecting and mixing synthetic sources.

## Impact Statement

This work studies how real and synthetic time-series corpora can be composed for pretraining time-series foundation models. Its primary impact is methodological: it cautions against treating synthetic data as a generic replacement for real data, or treating a synthetic-data recipe designed for one architecture as automatically transferable to another. Our results suggest that generator choice, mixture design, and real–synthetic ratios should be treated as explicit corpus design decisions and validated per model family.

This is especially relevant when real time-series data is limited, expensive, private, or difficult to share. In such settings, synthetic data can be a useful complement to availability-constrained real corpora, but poorly chosen generators or mixture ratios may introduce unrealistic dynamics, miss rare behaviours, or produce models that perform well on aggregate benchmarks while failing in specific domains. Similarly, real-only pretraining should not be assumed to be optimal under fixed budgets and practical data constraints.

Because time-series models are often used in consequential domains such as healthcare, energy, finance, and infrastructure, corpus composition choices can affect downstream reliability. We therefore recommend reporting pretraining corpus composition, validating performance across domains, and treating synthetic data as a configurable complement to real data rather than a universal substitute.

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

## A. Synthetic Generator Descriptions

Each of the eleven generators produces univariate series of length 1024. Hyperparameter sampling distributions for all generators are reported in Appendix B.

### ARIMA

The ARIMA generator samples a randomized SARIMA-like process with non-seasonal ARMA dynamics, optional ordinary integration, and optional seasonal lag structure (Box & Jenkins, 1970; Hamilton, 1994). It first simulates an ARMA$(p, q)$ process via

$$y_t = \sum_{j=1}^{p} \phi_j \, y_{t-j} + \varepsilon_t + \sum_{j=1}^{q} \theta_j \, \varepsilon_{t-j}, \qquad \varepsilon_t \sim \mathcal{N}(0, \, \sigma^2), \tag{1}$$

then optionally adds seasonal AR and MA terms at lag multiples of $m$. When enabled, the seasonal differencing order $D$ is applied before the ordinary integration order $d$. AR and MA coefficients are accepted only when the corresponding characteristic polynomial roots lie outside the unit circle. A burn-in of 500 steps is generated, after which the final length-$L$ segment is retained.

The corresponding sampling distributions are reported in Table 5.

### Chaotic

The Chaotic generator draws from two deterministic dynamical systems with equal probability: the Lorenz attractor (Lorenz, 1963) and the Mackey–Glass delay differential equation (Mackey & Glass, 1977). The Lorenz system is integrated via fourth-order Runge–Kutta,

$$\frac{dx}{dt} = \sigma(y - x), \quad \frac{dy}{dt} = x(\rho - z) - y, \quad \frac{dz}{dt} = xy - \beta_L z, \tag{2}$$

and one coordinate is extracted as the observed series. The Mackey–Glass equation is integrated via forward Euler,

$$\frac{dx}{dt} = \frac{\beta_M \, x(t - \tau)}{1 + x(t - \tau)^n} - \gamma_M \, x(t). \tag{3}$$

A burn-in of 2000 steps is discarded in both cases.

The corresponding sampling distributions are reported in Table 6.

### ETS

The ETS generator implements the full error–trend–seasonality state-space family (Hyndman et al., 2008), with error type $\in \{A, M\}$, trend $\in \{N, A, A_d\}$, and seasonality $\in \{N, A, M\}$ drawn uniformly. The observation and state-update equations for the multiplicative-error, damped-trend, multiplicative-seasonality variant are

$$\begin{aligned}
y_t &= (\ell_{t-1} + \phi b_{t-1}) \, s_{t-m} \, (1 + \varepsilon_t), \\
\ell_t &= \ell_{t-1} + \phi b_{t-1} + \alpha \, \varepsilon_t \, (\ell_{t-1} + \phi b_{t-1}), \\
b_t &= \phi \, b_{t-1} + \beta \, (\ell_t - \ell_{t-1}), \\
s_t &= s_{t-m} \, (1 + \gamma \, \varepsilon_t),
\end{aligned} \tag{4}$$

where $\varepsilon_t \sim \mathcal{N}(0, \sigma^2)$. Additive-error and additive-seasonality variants follow analogous recursions. A burn-in of 200 steps is discarded.

The corresponding sampling distributions are reported in Table 7.

### fBm

The fBm generator produces either fractional Brownian motion or its stationary increments (fractional Gaussian noise), governed by the Hurst exponent $H \in (0.1, 0.9)$ (Mandelbrot & Van Ness, 1968). The covariance structure is

$$C(i, j) = \tfrac{1}{2} \left( |i|^{2H} + |j|^{2H} - |i - j|^{2H} \right). \tag{5}$$

Samples are drawn via the Davies–Harte circulant-embedding method. The output is scaled by $s \sim \text{LogUniform}(0.1, 5.0)$.

The corresponding sampling distributions are reported in Table 8.

## GARCH

The GARCH generator draws from three conditional heteroskedasticity models: GARCH(1,1) (Bollerslev, 1986), GJR-GARCH (Glosten et al., 1993), and EGARCH (Nelson, 1991). All share the mean equation $r_t = \mu + \phi(r_{t-1} - \mu) + \varepsilon_t$ with $\varepsilon_t = \sigma_t z_t$. The variance equations are respectively

$$\sigma_t^2 = \omega + \alpha \varepsilon_{t-1}^2 + \beta \sigma_{t-1}^2, \tag{6}$$
$$\sigma_t^2 = \omega + \left(\alpha + \gamma \mathbf{1}[\varepsilon_{t-1} < 0]\right) \varepsilon_{t-1}^2 + \beta \sigma_{t-1}^2,$$
$$\log \sigma_t^2 = \omega + \alpha(|z_{t-1}| - \mathbb{E}|z|) + \gamma z_{t-1} + \beta \log \sigma_{t-1}^2.$$

Innovations $z_t$ are drawn from either $\mathcal{N}(0,1)$ or a standardised Student-$t$. A burn-in of 500 steps is discarded. With probability 0.5 the output is cumulatively summed to produce an integrated returns series.

The corresponding sampling distributions are reported in Table 9.

## KernelSynth

The KernelSynth generator draws samples from a Gaussian process prior (Ansari et al., 2024; Rasmussen & Williams, 2006),

$$y \sim \mathcal{GP}(m(\mathbf{x}), k(\mathbf{x}, \mathbf{x}')), \tag{7}$$

where the kernel $k$ is constructed by randomly composing 1–5 base kernels from a bank of 41, including RBF, Matérn ($\nu \in \{0.5, 1.5, 2.5\}$), rational quadratic, periodic (ExpSineSquared), dot-product, and white-noise kernels, combined via additions (probability 0.8) or products (probability 0.2). The mean function $m(\mathbf{x})$ is sampled with probability 0.5 from {linear, quadratic, sinusoidal, log-linear, random walk}.

The corresponding sampling distributions are reported in Table 10.

## SDE

The SDE generator implements a regime-switching Ornstein–Uhlenbeck process (Uhlenbeck & Ornstein, 1930; Hamilton, 1989), with $K \in \{2, 3, 4\}$ regimes governed by a Markov transition matrix $P$. Within regime $k$, the exact discrete-time transition (Gillespie, 1996) is

$$x_{t+1} = \mu_k + (x_t - \mu_k) e^{-\theta_k \Delta t} + \sigma_k \sqrt{\frac{1 - e^{-2\theta_k \Delta t}}{2\theta_k}} \, \eta_t, \quad \eta_t \sim \mathcal{N}(0,1), \tag{8}$$

with $\Delta t = 1$. Regime means $\{\mu_k\}$ are spaced by random signed offsets of $[1, 4]$; mean-reversion rates $\theta_k \in (0.05, 5.0)$ and noise scales $\sigma_k \in (0.05, 2.0)$ are sampled independently per regime.

The corresponding sampling distributions are reported in Table 11.

## StepFunction

The StepFunction generator produces piecewise-constant series (Moroshan et al., 2025) with $n_{\text{seg}} \sim \text{Geometric}(0.15) + 2$ segments whose lengths are drawn from a Dirichlet distribution. Segment levels follow one of three modes: uniform draws from $[-5, 5]$, a random walk with Gaussian increments, or cluster-assigned values. Transitions between segments are rendered as hard steps, linear ramps, or sigmoid blends over a window of width $\max(3, \lfloor U(0.01, 0.05) \cdot L \rfloor)$ timesteps. Gaussian noise is added with probability 0.7.

The corresponding sampling distributions are reported in Table 12.

## TimeSynth

The TimeSynth generator mixes 1–3 signal components drawn from a library of five types — sinusoidal, continuous autoregressive (CAR), NARMA, pseudoperiodic, and autoregressive — using the TimeSynth library (Maat et al., 2017).

Components are combined additively (probability 0.8) or multiplicatively, then a noise process drawn uniformly from a bank of 14 Gaussian and red-noise configurations is added. The CAR signal satisfies

$$dx_t = -\alpha\, x_t\, dt + \sigma\, dW_t, \tag{9}$$

and the NARMA signal follows a nonlinear autoregressive moving-average recursion of order $r \in \{5, \dots, 14\}$.

The corresponding sampling distributions are reported in Table 13.

**TSI**

The TSI generator constructs series by combining trend ($T$), seasonality ($S$), and irregularity ($N$) components (Bahrpeyma et al., 2021), with component types drawn uniformly from parametric families including a null option for each. Eight combination models are used; the fully multiplicative variant is

$$y_t = T_t\,(1 + 0.1\,S_t)\,(1 + 0.1\,N_t), \tag{10}$$

with additive and mixed forms following analogously. Trend types include linear, nonlinear, and none; seasonality types include sinusoidal, step, triangular, impulsive, and none; irregularity is drawn from fractional Gaussian noise, fractional Brownian motion, or white noise.

The corresponding sampling distributions are reported in Table 14.

**Waveform**

The Waveform generator produces mixtures of 1–3 non-smooth periodic signals (Moroshan et al., 2025), targeting asymmetric and discontinuous dynamics absent from GP-based generators. Each component is

$$w_t = A \cdot f(2\pi\nu\, t + \varphi), \tag{11}$$

where $f \in \{\text{sawtooth, square, triangle}\}$, amplitude $A \sim U(0.3, 3.0)$, frequency $\nu \sim U(1, 50)$, and phase $\varphi \sim U(0, 2\pi)$. An amplitude modulation envelope is applied with probability 0.3, a linear trend with probability 0.3, and additive Gaussian noise with probability 0.7.

The corresponding sampling distributions are reported in Table 15.

## B. Hyperparameter Distributions

Synthetic generators are instantiated by sampling from broad hyperparameter distributions rather than fixed configurations. These tables specify the parameter ranges used to construct the synthetic training corpora.

*Table 5.* ARIMA hyperparameter sampling distributions.

| Parameter | Sampling rule |
|---|---|
| $p$ | Categorical over $\{0, 1, 2, 3\}$ with weights $[0.1, 0.3, 0.4, 0.2]$; set $p{=}1$ if $p{=}q{=}0$ |
| $d$ | Categorical over $\{0, 1, 2\}$ with weights $[0.4, 0.4, 0.2]$ |
| $q$ | Categorical over $\{0, 1, 2, 3\}$ with weights $[0.1, 0.3, 0.4, 0.2]$ |
| $\sigma$ | LogUniform$(0.01, 2.0)$ |
| $\phi_j$ | $U(-0.8, 0.8)$, with stationarity enforced; fallback to $U(-0.3, 0.3)$ after 100 failed trials |
| $\theta_j$ | $U(-0.8, 0.8)$, with invertibility enforced; fallback to $U(-0.3, 0.3)$ after 100 failed trials |
| *Seasonal component, enabled with probability 0.3* | |
| $m$ | Categorical over $\{4, 7, 12, 24, 52\}$ |
| $P$ | Categorical over $\{0, 1\}$ |
| $Q$ | Categorical over $\{0, 1\}$; seasonal innovations use $\mathcal{N}(0, 0.5\sigma)$ |
| $D$ | Categorical over $\{0, 1\}$ |

*Table 6.* Chaotic hyperparameter sampling distributions.

| Parameter | Sampling rule |
|---|---|
| system | Uniform over {Lorenz, Mackey–Glass} |
| *Lorenz system* | |
| $\sigma$ | $U(8, 12)$ |
| $\rho$ | $U(24, 30)$ |
| $\beta_L$ | $U(2.0, 3.5)$ |
| $\Delta t$ | $U(0.005, 0.02)$; RK4 integration step size |
| $x_0$ | $U(-1, 1)^3$ |
| coordinate | Uniform over {0, 1, 2}; extracted observed axis |
| *Mackey–Glass system* | |
| $\tau$ | $U(15, 30)$ |
| $n$ | Uniform over {8, 9, 10, 11, 12} |
| $\beta_M$ | $U(0.15, 0.25)$ |
| $\gamma_M$ | $U(0.05, 0.15)$ |
| $\Delta t$ | $U(0.5, 2.0)$; Euler integration step size |
| $x_{\text{init}}$ | $U(0.8, 1.0)$; history initialisation |

*Table 7.* ETS hyperparameter sampling distributions.

| Parameter | Sampling rule |
|---|---|
| error type | Uniform over {A, M} |
| trend type | Uniform over {N, A, A$_d$} |
| season type | Uniform over {N, A, M} |
| $m$ | Uniform over {4, 7, 12, 24, 52} if season type is N |
| $\alpha$ | $U(0.01, 0.3)$ |
| $\beta$ | $U(0.001, 0.1)$ if trend type is not N |
| $\gamma$ | $U(0.001, 0.15)$ if season type is not N |
| $\phi$ | $U(0.8, 0.98)$ if trend type is A$_d$ |
| $\sigma_{\text{M}}$ | $U(0.005, 0.05)$ if error type is M |
| $\sigma_{\text{A}}$ | $\text{LogUniform}(0.01, 0.3)$ if error type is A |
| $\ell_0$ | $U(10, 50)$; initial level |
| $b_0$ | $U(-0.1, 0.1)$ if trend type is not N |
| $s_0$ additive | $\mathcal{N}(0, 0.05\ell_0)$; mean-centered initial seasonal states |
| $s_0$ multiplicative | $U(0.85, 1.15)$; normalised to mean 1 |

*Table 8.* fBm hyperparameter sampling distributions.

| Parameter | Sampling rule |
|---|---|
| $H$ | $U(0.1, 0.9)$; Hurst exponent |
| output type | Uniform over {fBm, fGn} |
| $s$ | $\text{LogUniform}(0.1, 5.0)$; output scale |

*Table 9.* GARCH hyperparameter sampling distributions.

| Parameter | Sampling rule |
|---|---|
| model | Uniform over {GARCH, GJR, EGARCH} |
| innovations | Uniform over $\{\mathcal{N}, t\}$ |
| mean type | Uniform over {zero, const, AR1} |
| $\nu$ | $U(3, 10)$ if innovations are Student-$t$ |
| $\mu$ | $\mathcal{N}(0, 0.05)$ if mean type is const |
| $\phi$ | $U(-0.3, 0.3)$ if mean type is AR1 |
| $\sigma_{\mathrm{unc}}$ | $\mathrm{LogUniform}(0.005, 0.5)$; unconditional volatility used to derive $\omega$ |
| cumsum | $\mathrm{Bernoulli}(0.5)$; cumulatively sum output to form an integrated series |
| *GARCH(1,1)* | |
| persistence | $U(0.70, 0.97)$; $\alpha + \beta$ |
| $\alpha$ share | $U(0.02, 0.25)$; $\alpha = \mathrm{persistence} \times \mathrm{share}$ |
| *GJR-GARCH* | |
| persistence | $U(0.70, 0.97)$ |
| $\alpha$ share | $U(0.02, 0.20)$ |
| $\gamma$ | $U(0.01, 0.15)$; leverage effect |
| *EGARCH* | |
| $\beta$ | $U(0.70, 0.97)$ |
| $\alpha$ | $U(0.05, 0.25)$ |
| $\gamma$ | $U(-0.20, 0.20)$; asymmetry parameter |

*Table 10.* KernelSynth hyperparameter sampling distributions.

| Parameter | Sampling rule |
|---|---|
| number of kernels | Uniform over $\{1, 2, 3, 4, 5\}$ |
| combination operator | Categorical over $\{+, \times\}$ with weights $[0.8, 0.2]$; applied pairwise |
| mean flag | $\mathrm{Bernoulli}(0.5)$ |
| mean type | Uniform over {linear, quadratic, sinusoidal, log-linear, RW} if mean flag is enabled |
| *Linear mean:* $at + b$ | |
| $a$ | $U(-3, 3)$ |
| $b$ | $U(-2, 2)$ |
| *Quadratic mean:* $at^2 + bt + c$ | |
| $a, b$ | $U(-2, 2)$ |
| $c$ | $U(-1, 1)$ |
| *Sinusoidal mean:* $A \sin(2\pi f t + \varphi)$ | |
| $A$ | $U(0.5, 3)$ |
| $f$ | $U(1, 8)$ |
| $\varphi$ | $U(0, 2\pi)$ |
| *Log-linear mean:* $a \log(1 + bt) + c$ | |
| $a$ | $U(-3, 3)$ |
| $b$ | $U(1, 20)$ |
| $c$ | $U(-1, 1)$ |
| *Random-walk mean* | |
| step std | $U(0.01, 0.1)$; cumulative sum |

*Table 11.* SDE hyperparameter sampling distributions.

| Parameter | Sampling rule |
|---|---|
| $K$ | Categorical over $\{2, 3, 4\}$ with weights $[0.5, 0.3, 0.2]$; number of regimes |
| $\mu_0$ | $U(-3.0, 3.0)$; base mean |
| $\Delta\mu_k$ | $U(1.0, 4.0)$ with random sign; offset per regime |
| $\theta_k$ | $U(0.05, 5.0)$; mean-reversion rate per regime |
| $\sigma_k$ | $U(0.05, 2.0)$; noise scale per regime |
| diagonal concentration | $U(15.0, 50.0)$; Dirichlet parameter for transition matrix $P$ |
| initial regime | Uniform over $\{0, \ldots, K-1\}$ |

*Table 12.* StepFunction hyperparameter sampling distributions.

| Parameter | Sampling rule |
|---|---|
| $n_{\text{seg}}$ | Geometric$(0.15) + 2$, capped at $\max(3, L/4)$ |
| level mode | Uniform over {uniform, random walk, clustered} |
| transition type | Categorical over {hard, ramp, sigmoid} with weights $[0.5, 0.3, 0.2]$ |
| Dirichlet concentration | $U(0.5, 2.0)$; segment length proportions |
| *Uniform levels* | |
| level | $U(-5, 5)$ per segment |
| *Random-walk levels* | |
| start | $U(-3, 3)$ |
| increment std | $U(0.3, 2.0)$ |
| *Clustered levels* | |
| $n_{\text{clusters}}$ | Uniform over $\{2, \ldots, \min(5, n_{\text{seg}})\}$ |
| cluster centers | $U(-5, 5)$ |
| jitter | $\mathcal{N}(0, 0.1)$ per segment |
| *Transitions, ramp/sigmoid only* | |
| window width | $U(0.01, 0.05) \times L$, with minimum width of 3 steps |
| sigmoid steepness | $U(5, 15)$ |
| *Noise* | |
| noise flag | Bernoulli$(0.7)$ |
| noise std | $U(0.01, 0.3)$ if noise is enabled |

*Table 13.* TimeSynth hyperparameter sampling distributions.

| Parameter | Sampling rule |
|---|---|
| $n_{\text{signals}}$ | Uniform over $\{1, 2, 3\}$ |
| signal type | Categorical over {sin, CAR, NARMA, PP, AR} with weights $[0.20, 0.25, 0.20, 0.15, 0.20]$ |
| weight $w$ | $U(0.4, 1.2)$ per component |
| combination | Categorical over $\{+, \times\}$ with weights $[0.8, 0.2]$ |
| irregular probability | $U(0.1, 0.3)$; forced regular for AR/NARMA components |
| keep percentage | $U(80, 95)$ if irregular sampling is enabled |
| *Sinusoidal component* | |
| period $p$ | Uniform over $\{4, \ldots, 127\}$ |
| amplitude | $U(0.5, 3.0)$ |
| *CAR component* | |
| $\alpha$ | $U(0.1, 0.95)$; decay parameter |
| $\sigma$ | $U(0.1, 0.8)$ |
| start value | $U(-0.5, 0.5)$ |
| *NARMA component* | |
| order $r$ | Uniform over $\{5, \ldots, 14\}$ |
| $c_1$ | $U(0.6, 0.9)$ |
| $c_2$ | $U(0.02, 0.08)$ |
| $c_3$ | $U(1.0, 2.0)$ |
| $c_4$ | $U(0.05, 0.15)$ |
| *Pseudoperiodic component* | |
| period $p$ | Uniform over $\{6, \ldots, 101\}$ |
| amplitude | $U(0.5, 2.5)$ |
| ampSD | $U(0.05, 0.25)$ |
| freqSD | $U(0.01, 0.15) \times p$ |
| *Autoregressive component* | |
| order $p$ | Uniform over $\{1, 2, 3\}$ |
| $\phi_j$ | $U(-0.6, 0.6)$ with $\sum_j |\phi_j| \leq 0.9$ |
| $\sigma$ | $U(0.1, 0.7)$ |

*Table 14.* TSI hyperparameter sampling distributions.

| Parameter | Sampling rule |
|---|---|
| trend type | Uniform over {linear, nonlinear, none} |
| season type | Uniform over {sinusoidal, step, triangular, impulsive, none} |
| irregularity type | Uniform over {fGn, fBm, white} |
| combination model | Uniform over $\{1, \ldots, 8\}$ |
| *Linear trend: $ct + d$* | |
| $c$ | $U(-2, 2)$ |
| $d$ | $U(-1, 1)$ |
| *Nonlinear trend: $(at + b)\sin(2\pi t) + ct + d$* | |
| $a, b$ | $U(-1, 1)$ |
| $c$ | $U(-2, 2)$ |
| $d$ | $U(-1, 1)$ |
| *Seasonality, sinusoidal/step/triangular/impulsive* | |
| period | Uniform over $\{4, \ldots, \max(7, L/8) - 1\}$ |
| amplitude | $U(0.5, 2.0)$ |
| phase | $U(0, 2\pi)$ for sinusoidal seasonality |
| *Irregularity* | |
| $H$ | $U(0.3, 0.7)$ for fGn, using AR approximation |
| scale | $U(0.5, 0.9)$ for fBm, using normalised cumulative sum |
| std | Fixed at 0.2 for white noise |

*Table 15.* Waveform hyperparameter sampling distributions.

| Parameter | Sampling rule |
|---|---|
| $n_{\text{waves}}$ | Uniform over $\{1, 2, 3\}$ |
| waveform type | Uniform over {sawtooth, square, triangle} per component |
| $A$ | $U(0.3, 3.0)$; amplitude per component |
| $\nu$ | $U(1, 50)$; cycles over $[0, 1]$ |
| $\varphi$ | $U(0, 2\pi)$; phase per component |
| duty cycle | $U(0.2, 0.8)$ for square waves |
| sawtooth width | Uniform over $\{0.0, 1.0\}$ for sawtooth waves |
| *Optional modifications* | |
| AM envelope | Bernoulli(0.3) |
| AM frequency | $U(0.5, 5.0)$ if amplitude modulation is enabled |
| noise flag | Bernoulli(0.7) |
| noise std | $U(0.01, 0.3)$ if noise is enabled |
| linear trend | Bernoulli(0.3) |
| trend slope | $U(-2, 2)$ if linear trend is enabled |

## C. Training Corpus Construction

Table 16 summarises the pretraining corpora used in the downstream experiments. All conditions use 1M univariate windows of length 1024. For real–synthetic mixture conditions, three model initialisations are trained per corpus; all single-generator and Mixed11 conditions use a single run.

## D. Real Reference Corpus: Sampling Details

The real reference corpus is drawn from the GIFT-Eval pretraining dataset (Aksu et al., 2024). We sample 1,000,000 univariate windows of length 1024, stratified by frequency and domain where possible. Because available long-context real series are unevenly distributed across domains, this corpus is availability-constrained rather than uniformly balanced.

ALLOCATION CONSTRAINTS

We apply three constraints when constructing the real reference corpus. First, datasets with mean series length below 1024 are excluded so that eligible source series can provide length-1024 windows. Second, viable frequency–domain strata receive minimum allocations when sufficient supply is available. Third, per-dataset caps are applied within strata to reduce dominance by very large sources.

*Table 16.* Training corpus manifest. All conditions use 1M windows of length 1024.

| Condition | Real windows | Synthetic windows | Synthetic source |
|---|---|---|---|
| ARIMA | 0 | 1,000,000 | ARIMA |
| Chaotic | 0 | 1,000,000 | Chaotic |
| ETS | 0 | 1,000,000 | ETS |
| fBm | 0 | 1,000,000 | fBm |
| GARCH | 0 | 1,000,000 | GARCH |
| KernelSynth | 0 | 1,000,000 | KernelSynth |
| SDE | 0 | 1,000,000 | SDE |
| StepFunction | 0 | 1,000,000 | StepFunction |
| TimeSynth | 0 | 1,000,000 | TimeSynth |
| TSI | 0 | 1,000,000 | TSI |
| Waveform | 0 | 1,000,000 | Waveform |
| Mixed11 | 0 | 1,000,000 | equal mixture of all 11 generators |
| Real reference | 1,000,000 | 0 | – |
| 75-25 real-synthetic | 750,000 | 250,000 | Mixed11 synthetic source |
| 50-50 real-synthetic | 500,000 | 500,000 | Mixed11 synthetic source |
| 25-75 real-synthetic | 250,000 | 750,000 | Mixed11 synthetic source |

Interior missing values are forward- and backward-filled; trailing missing values are trimmed. Windows with fewer than 128 valid observations after trimming are discarded. When a dataset cannot supply its allocated number of valid windows, shortfalls are first retried within the same dataset where possible and then filled from the remaining available datasets.

## SAMPLING AUDIT

Table 17 summarises the final sampling audit for the real reference corpus. The final sample contains 1,000,000 windows across 126 datasets and 19 viable frequency–domain strata.

*Table 17.* Real reference sampling audit computed from final metadata.

| Quantity | Value |
|---|---|
| Final sampled windows | 1,000,000 |
| Datasets contributing windows | 126 |
| Viable frequency–domain strata | 19 |
| Backfilled windows | 2,057 |
| Windows with full valid length 1024 | 996,256 |
| Windows with valid length $< 1024$ after trimming | 3,744 |

## TRAIN–TEST LEAKAGE AVOIDANCE

The real reference corpus is sampled from the GIFT-Eval pretraining pool and excludes downstream evaluation windows used for zero-shot testing. The sampled real-reference corpus is fixed across model seeds. Therefore, seed-level variation reflects model initialisation and training stochasticity rather than resampling of the real corpus.

## STRATUM COVERAGE

Table 18 summarises allocation across viable frequency–domain strata in the final sampled metadata. The final sample reflects the uneven availability of long-context real-world series after applying the viability and allocation constraints.

*Table 18.* Allocation across viable frequency–domain strata in the real reference corpus, computed from final sampled metadata.

| Freq | Domain | Datasets | Windows | Share |
|------|--------|----------|---------|-------|
| H | Energy | 20 | 558,594 | 55.9% |
| 5T | Transport | 11 | 160,269 | 16.0% |
| H | Climate | 31 | 76,169 | 7.6% |
| 6H | Climate | 33 | 69,987 | 7.0% |
| 5T | CloudOps | 3 | 51,774 | 5.2% |
| D | Web | 1 | 13,155 | 1.3% |
| 15T | Transport | 3 | 9,726 | 1.0% |
| 30T | Energy | 3 | 9,726 | 1.0% |
| D | Climate | 3 | 9,726 | 1.0% |
| D | Econ/Fin | 2 | 9,726 | 1.0% |
| D | Sales | 2 | 9,726 | 1.0% |
| H | Transport | 3 | 8,472 | 0.8% |
| T | Energy | 3 | 6,728 | 0.7% |
| H | Nature | 2 | 3,592 | 0.4% |
| W | Healthcare | 1 | 1,258 | 0.1% |
| 10T | Energy | 1 | 1,072 | 0.1% |
| 30T | Transport | 1 | 276 | $< 0.1\%$ |
| 4S | Energy | 2 | 16 | $< 0.1\%$ |
| D | Nature | 1 | 8 | $< 0.1\%$ |
| **Total** | | **126** | **1,000,000** | **100%** |

Although the allocation procedure imposes floors and caps, the final corpus remains concentrated because several smaller strata exhaust their available supply, while large long-series datasets retain substantial headroom. We therefore treat the real reference as an availability-constrained reference corpus, not as a uniformly balanced estimate of all time-series domains.

## E. TSTR Training Configuration

Table 19 reports the shared training configuration used for Chronos-T5-Mini and Moirai-Small. Both models are trained from random initialisation without pretrained weights. For each condition, the sampled pretraining corpus is fixed across seeds; only the model initialisation and training seed vary. Window length denotes the sampled source windows used to construct the pretraining corpora, while context length denotes the model context used during training.

*Table 19.* Training configuration for downstream pretraining experiments.

| Hyperparameter | Chronos-T5-Mini | Moirai-Small |
|----------------|-----------------|--------------|
| Parameters | 20M | 13.8M |
| Context length | 512 | 512 |
| Optimiser | AdamW | AdamW |
| Learning rate | $10^{-3}$ | $10^{-3}$ |
| LR schedule | Linear decay | Linear decay |
| Dataloader batch size | 96 | 96 |
| Training steps | 50,000 | 50,000 |
| Model seeds | 42, 43, 44 | 42, 43, 44 |
| Pretrained weights | None | None |
| Training corpus | 1M windows | 1M windows |
| Window length | 1024 | 1024 |

The configured dataloader batch size is reported rather than a hardware-specific effective batch size. Evaluation uses the trained checkpoints zero-shot; no downstream fine-tuning is performed on GIFT-Eval evaluation datasets.

## F. Evaluation and Aggregation Protocol

All trained checkpoints are evaluated zero-shot on the 28-dataset GIFT-Eval benchmark. We report normalized CRPS and normalized MASE, matching the metrics used in the main text. Both metrics are normalized against the seasonal-naive baseline, so values below 1 indicate improvement over that baseline and lower values are better.

For each model, condition, seed, dataset, and horizon, we first compute the normalized metric value. We treat each dataset–horizon pair as an evaluation task. Aggregate scores in the main text are then computed as geometric means across these task-level values. For metric values $m_1, \ldots, m_N$ across $N$ evaluation tasks, the aggregate is

$$\overline{m}_{\text{geo}} = \exp\left(\frac{1}{N}\sum_{i=1}^{N}\log m_i\right). \tag{12}$$

We report the mean and standard deviation of these aggregate scores over model seeds. The sampled pretraining corpora are fixed across seeds, so the reported seed standard deviations reflect model initialisation and training stochasticity rather than corpus-resampling variance.

Domain-level tables are computed by grouping GIFT-Eval tasks by thematic domain and forecast horizon. Missing domain–horizon cells are reported as "–" when no corresponding evaluation dataset is available.

## G. Bootstrap Confidence Intervals

We compute paired bootstrap confidence intervals for the main condition comparisons. For each model, comparison, and metric, we first aggregate scores by evaluation task and condition. We then compute paired log differences between the two conditions:

$$d_i = \log m_i^{(a)} - \log m_i^{(b)},$$

where lower metric values are better. A negative value therefore means condition $a$ outperforms condition $b$.

We resample the paired task-level differences with replacement for 10,000 bootstrap replicates using random seed 42. Relative deltas are reported as

$$\exp\left(\frac{1}{N}\sum_{i=1}^{N}d_i\right) - 1,$$

with 95% percentile confidence intervals computed from the bootstrap distribution. We also report $p(a < b)$, the fraction of bootstrap replicates in which the first condition has lower error, and the empirical task-level win rate, defined as the fraction of evaluation tasks on which the first condition achieves lower error, as descriptive statistics.

The paired bootstrap quantifies uncertainty over sampled evaluation tasks, not uncertainty over pretraining runs. The single-generator comparisons use the seed-42 generator sweep. The composition comparisons pool the three available composition seeds, 42, 43, and 44, giving 291 paired seed–task observations. These intervals should therefore be interpreted as task-level evidence for the reported comparisons, not as definitive estimates of training-run uncertainty.

### Best and Second-Best Single-Generator Corpora

Table 20 compares the best seed-42 single-generator corpus against the second-best single-generator corpus, Mixed11, and the real-reference corpus. KernelSynth is the best single-generator corpus by average CRPS/MASE rank for both architectures; the second-best generator is ETS for Moirai-Small and SDE for Chronos-T5-Mini. Negative relative deltas indicate that the first condition has lower error.

For Moirai-Small, KernelSynth improves over ETS and Real Ref, while its aggregate comparison against Mixed11 is effectively tied for CRPS and slightly favours KernelSynth for MASE. For Chronos-T5-Mini, KernelSynth is not clearly better than SDE, is worse than Mixed11 on MASE, and is clearly worse than the real-reference corpus.

### Best Real–Synthetic Mixture versus Pure-Source Baselines

Table 21 compares the best-performing real–synthetic mixture for each model against the second-best mixture ratio, the pure synthetic Mixed11 corpus, and the real reference corpus. Negative relative deltas indicate that the selected real–synthetic mixture has lower error.

*Table 20.* Seed-42 paired bootstrap comparisons for the best single-generator corpus against the second-best single-generator corpus, Mixed11, and the real-reference corpus. Relative deltas are percentages on the normalized metric scale; negative values indicate that the first condition has lower error. $p(a < b)$ denotes the fraction of bootstrap replicates in which the first condition has lower error.

| Model | Comparison | Metric | $N$ | Rel. $\Delta$ | 95% CI | $p(a < b)$ | Win rate |
|---|---|---|---|---|---|---|---|
| Moirai-Small | KernelSynth vs ETS | CRPS | 97 | -10.6% | [-14.9%, -6.3%] | 1.00 | 0.62 |
| Moirai-Small | KernelSynth vs ETS | MASE | 97 | -9.1% | [-13.3%, -4.9%] | 1.00 | 0.62 |
| Moirai-Small | KernelSynth vs Mixed11 | CRPS | 97 | -0.2% | [-3.7%, 3.3%] | 0.54 | 0.48 |
| Moirai-Small | KernelSynth vs Mixed11 | MASE | 97 | -1.9% | [-5.5%, 1.4%] | 0.85 | 0.47 |
| Moirai-Small | KernelSynth vs Real Ref | CRPS | 97 | -9.9% | [-15.4%, -4.2%] | 1.00 | 0.62 |
| Moirai-Small | KernelSynth vs Real Ref | MASE | 97 | -8.7% | [-13.6%, -3.7%] | 1.00 | 0.62 |
| Chronos-T5-Mini | KernelSynth vs SDE | CRPS | 97 | -4.5% | [-10.8%, 2.1%] | 0.91 | 0.59 |
| Chronos-T5-Mini | KernelSynth vs SDE | MASE | 97 | 0.6% | [-6.7%, 8.0%] | 0.42 | 0.48 |
| Chronos-T5-Mini | KernelSynth vs Mixed11 | CRPS | 97 | 3.3% | [-1.2%, 8.2%] | 0.08 | 0.42 |
| Chronos-T5-Mini | KernelSynth vs Mixed11 | MASE | 97 | 10.2% | [5.5%, 15.2%] | 0.00 | 0.34 |
| Chronos-T5-Mini | KernelSynth vs Real Ref | CRPS | 97 | 18.3% | [10.8%, 26.8%] | 0.00 | 0.32 |
| Chronos-T5-Mini | KernelSynth vs Real Ref | MASE | 97 | 21.6% | [13.9%, 30.2%] | 0.00 | 0.29 |

These comparisons assess whether the selected real–synthetic mixture improves over nearby mixture ratios and pure-source baselines under paired task-level resampling. For Moirai-Small, the 75–25 real–synthetic mixture improves over 50–50, Mixed11, and Real Ref on both metrics. For Chronos-T5-Mini, the 50–50 mixture improves over 25–75 and Mixed11, and is competitive with Real Ref: the CRPS interval overlaps zero, while MASE favours 50–50.

*Table 21.* Three-seed paired bootstrap comparisons for the selected real–synthetic mixture against the second-best mixture ratio and pure-source baselines. Relative deltas are percentages on the normalized metric scale; negative values indicate that the first condition has lower error. $p(a < b)$ denotes the fraction of bootstrap replicates in which the first condition has lower error.

| Model | Comparison | Metric | $N$ | Rel. $\Delta$ | 95% CI | $p(a < b)$ | Win rate |
|---|---|---|---|---|---|---|---|
| Moirai-Small | 75-25 vs 50-50 | CRPS | 291 | -3.6% | [-5.3%, -1.8%] | 1.00 | 0.59 |
| Moirai-Small | 75-25 vs 50-50 | MASE | 291 | -2.8% | [-4.4%, -1.1%] | 1.00 | 0.64 |
| Moirai-Small | 75-25 vs Mixed11 | CRPS | 291 | -6.6% | [-8.4%, -4.7%] | 1.00 | 0.71 |
| Moirai-Small | 75-25 vs Mixed11 | MASE | 291 | -7.0% | [-8.9%, -5.2%] | 1.00 | 0.72 |
| Moirai-Small | 75-25 vs Real Ref | CRPS | 291 | -17.5% | [-20.0%, -15.0%] | 1.00 | 0.88 |
| Moirai-Small | 75-25 vs Real Ref | MASE | 291 | -16.0% | [-18.3%, -13.8%] | 1.00 | 0.88 |
| Chronos-T5-Mini | 50-50 vs 25-75 | CRPS | 291 | -1.5% | [-2.9%, -0.1%] | 0.98 | 0.57 |
| Chronos-T5-Mini | 50-50 vs 25-75 | MASE | 291 | -1.9% | [-2.9%, -0.9%] | 1.00 | 0.59 |
| Chronos-T5-Mini | 50-50 vs Mixed11 | CRPS | 291 | -13.2% | [-15.7%, -10.8%] | 1.00 | 0.80 |
| Chronos-T5-Mini | 50-50 vs Mixed11 | MASE | 291 | -12.5% | [-14.7%, -10.3%] | 1.00 | 0.81 |
| Chronos-T5-Mini | 50-50 vs Real Ref | CRPS | 291 | -1.0% | [-3.6%, 1.7%] | 0.77 | 0.51 |
| Chronos-T5-Mini | 50-50 vs Real Ref | MASE | 291 | -3.1% | [-5.3%, -0.8%] | 0.99 | 0.61 |

# H. Extended TSTR Results

This section reports domain-level zero-shot results that qualify the aggregate results in Section 3. Tables are grouped by model family and corpus setting. These results are intended as evidence for domain- and horizon-level heterogeneity, not as additional headline claims.

For the real–synthetic mixture sweep, we report mean ± standard deviation across seeds 42, 43, and 44. Lower values are better throughout.

**Moirai-Small**

*Table 22.* Three-seed real–synthetic mixture domain-level performance for Moirai-Small. Scores are geometric means across evaluation tasks within each domain and horizon, reported as mean ± standard deviation across seeds 42, 43, and 44. Condition labels denote real–synthetic window ratios, with Real Ref and Mixed11 as pure-source endpoints. Lower is better; **bold** marks the best value per domain, horizon, and metric.

| Domain | R-S Ratio | Short | | Medium | | Long | |
|---|---|---|---|---|---|---|---|
| | | CRPS | MASE | CRPS | MASE | CRPS | MASE |
| Econ/Fin | Real Ref | 1.18 ± 0.03 | 1.51 ± 0.02 | – | – | – | – |
| | 75-25 | 1.04 ± 0.13 | 1.18 ± 0.19 | – | – | – | – |
| | 50-50 | **0.97 ± 0.03** | **1.06 ± 0.02** | – | – | – | – |
| | 25-75 | 0.99 ± 0.06 | 1.09 ± 0.03 | – | – | – | – |
| | Mixed11 | 1.10 ± 0.12 | 1.24 ± 0.15 | – | – | – | – |
| Energy | Real Ref | 0.95 ± 0.00 | 1.16 ± 0.01 | 1.15 ± 0.02 | 1.55 ± 0.02 | 1.31 ± 0.07 | 1.90 ± 0.12 |
| | 75-25 | 0.80 ± 0.02 | 0.99 ± 0.01 | **0.80 ± 0.02** | **1.15 ± 0.03** | **0.92 ± 0.09** | **1.41 ± 0.10** |
| | 50-50 | **0.80 ± 0.01** | **0.99 ± 0.01** | 0.92 ± 0.07 | 1.29 ± 0.10 | 1.08 ± 0.08 | 1.60 ± 0.11 |
| | 25-75 | 0.83 ± 0.01 | 1.03 ± 0.01 | 1.23 ± 0.06 | 1.64 ± 0.07 | 1.31 ± 0.02 | 1.92 ± 0.08 |
| | Mixed11 | 0.84 ± 0.01 | 1.04 ± 0.00 | 0.94 ± 0.03 | 1.37 ± 0.04 | 1.02 ± 0.08 | 1.60 ± 0.11 |
| Healthcare | Real Ref | 0.79 ± 0.01 | 0.95 ± 0.01 | – | – | – | – |
| | 75-25 | **0.55 ± 0.02** | **0.71 ± 0.04** | – | – | – | – |
| | 50-50 | 0.56 ± 0.01 | 0.72 ± 0.02 | – | – | – | – |
| | 25-75 | 0.61 ± 0.02 | 0.77 ± 0.04 | – | – | – | – |
| | Mixed11 | 0.57 ± 0.02 | 0.73 ± 0.02 | – | – | – | – |
| Nature | Real Ref | 0.56 ± 0.01 | 0.84 ± 0.01 | 0.42 ± 0.02 | 1.19 ± 0.04 | 0.41 ± 0.04 | 1.13 ± 0.03 |
| | 75-25 | 0.53 ± 0.01 | 0.80 ± 0.03 | **0.35 ± 0.01** | **1.00 ± 0.01** | **0.33 ± 0.02** | **1.01 ± 0.02** |
| | 50-50 | **0.52 ± 0.00** | **0.79 ± 0.00** | 0.35 ± 0.01 | 1.03 ± 0.03 | 0.36 ± 0.02 | 1.06 ± 0.03 |
| | 25-75 | 0.53 ± 0.01 | 0.80 ± 0.01 | 0.42 ± 0.04 | 1.18 ± 0.03 | 0.42 ± 0.04 | 1.16 ± 0.04 |
| | Mixed11 | 0.54 ± 0.00 | 0.80 ± 0.01 | 0.39 ± 0.01 | 1.04 ± 0.05 | 0.38 ± 0.02 | 1.07 ± 0.07 |
| Sales | Real Ref | 0.46 ± 0.00 | 0.75 ± 0.00 | – | – | – | – |
| | 75-25 | **0.44 ± 0.00** | **0.73 ± 0.00** | – | – | – | – |
| | 50-50 | 0.45 ± 0.00 | 0.73 ± 0.01 | – | – | – | – |
| | 25-75 | 0.45 ± 0.00 | 0.73 ± 0.00 | – | – | – | – |
| | Mixed11 | 0.49 ± 0.00 | 0.82 ± 0.01 | – | – | – | – |
| Transport | Real Ref | 0.66 ± 0.01 | 0.78 ± 0.02 | 0.75 ± 0.02 | 1.00 ± 0.01 | 0.81 ± 0.02 | 1.17 ± 0.03 |
| | 75-25 | 0.63 ± 0.01 | 0.74 ± 0.01 | **0.62 ± 0.01** | **0.84 ± 0.01** | **0.67 ± 0.06** | **1.00 ± 0.07** |
| | 50-50 | **0.62 ± 0.00** | **0.73 ± 0.00** | 0.67 ± 0.05 | 0.90 ± 0.07 | 0.73 ± 0.04 | 1.08 ± 0.06 |
| | 25-75 | 0.63 ± 0.01 | 0.74 ± 0.01 | 0.81 ± 0.06 | 1.06 ± 0.08 | 0.82 ± 0.03 | 1.19 ± 0.06 |
| | Mixed11 | 0.68 ± 0.00 | 0.81 ± 0.00 | 0.75 ± 0.03 | 1.00 ± 0.03 | 0.73 ± 0.04 | 1.07 ± 0.06 |
| Web/CloudOps | Real Ref | 0.79 ± 0.05 | 1.17 ± 0.07 | 1.02 ± 0.03 | 1.41 ± 0.05 | 0.99 ± 0.05 | 1.35 ± 0.06 |
| | 75-25 | 0.56 ± 0.03 | **0.86 ± 0.04** | **0.89 ± 0.00** | **1.24 ± 0.01** | 0.94 ± 0.03 | **1.30 ± 0.05** |
| | 50-50 | 0.60 ± 0.02 | 0.89 ± 0.02 | 0.93 ± 0.05 | 1.32 ± 0.06 | **0.93 ± 0.03** | 1.32 ± 0.07 |
| | 25-75 | 0.64 ± 0.03 | 0.96 ± 0.04 | 0.98 ± 0.07 | 1.39 ± 0.11 | 0.96 ± 0.05 | 1.43 ± 0.15 |
| | Mixed11 | **0.56 ± 0.02** | 0.87 ± 0.01 | 0.91 ± 0.03 | 1.33 ± 0.02 | 0.95 ± 0.03 | 1.42 ± 0.07 |

**Chronos-T5-Mini**

*Table 23.* Three-seed real–synthetic mixture domain-level performance for Chronos-T5-Mini. Scores are geometric means across evaluation tasks within each domain and horizon, reported as mean $\pm$ standard deviation across seeds 42, 43, and 44. Condition labels denote real–synthetic window ratios, with Real Ref and Mixed11 as pure-source endpoints. Lower is better; **bold** marks the best value per domain, horizon, and metric.

| Domain | R-S Ratio | Short | | Medium | | Long | |
|---|---|---|---|---|---|---|---|
| | | CRPS | MASE | CRPS | MASE | CRPS | MASE |
| Econ/Fin | Real Ref | $0.89 \pm 0.01$ | $1.02 \pm 0.01$ | – | – | – | – |
| | 75-25 | $0.88 \pm 0.03$ | $0.88 \pm 0.01$ | – | – | – | – |
| | 50-50 | $0.91 \pm 0.03$ | $\mathbf{0.87 \pm 0.01}$ | – | – | – | – |
| | 25-75 | $\mathbf{0.87 \pm 0.03}$ | $0.89 \pm 0.02$ | – | – | – | – |
| | Mixed11 | $0.97 \pm 0.02$ | $0.98 \pm 0.02$ | – | – | – | – |
| Energy | Real Ref | $0.91 \pm 0.01$ | $1.14 \pm 0.01$ | $1.22 \pm 0.03$ | $\mathbf{1.46 \pm 0.02}$ | $\mathbf{1.12 \pm 0.03}$ | $\mathbf{1.44 \pm 0.03}$ |
| | 75-25 | $0.84 \pm 0.01$ | $1.04 \pm 0.02$ | $1.28 \pm 0.01$ | $1.57 \pm 0.03$ | $1.25 \pm 0.05$ | $1.64 \pm 0.06$ |
| | 50-50 | $\mathbf{0.81 \pm 0.01}$ | $\mathbf{0.99 \pm 0.01}$ | $\mathbf{1.21 \pm 0.01}$ | $1.49 \pm 0.02$ | $1.23 \pm 0.06$ | $1.58 \pm 0.08$ |
| | 25-75 | $0.84 \pm 0.01$ | $1.05 \pm 0.01$ | $1.23 \pm 0.02$ | $1.53 \pm 0.02$ | $1.21 \pm 0.02$ | $1.60 \pm 0.02$ |
| | Mixed11 | $0.92 \pm 0.02$ | $1.13 \pm 0.02$ | $1.62 \pm 0.01$ | $1.92 \pm 0.02$ | $1.53 \pm 0.06$ | $1.94 \pm 0.07$ |
| Healthcare | Real Ref | $0.85 \pm 0.03$ | $0.89 \pm 0.03$ | – | – | – | – |
| | 75-25 | $0.65 \pm 0.04$ | $0.73 \pm 0.03$ | – | – | – | – |
| | 50-50 | $\mathbf{0.61 \pm 0.01}$ | $\mathbf{0.69 \pm 0.02}$ | – | – | – | – |
| | 25-75 | $0.63 \pm 0.03$ | $0.71 \pm 0.04$ | – | – | – | – |
| | Mixed11 | $0.67 \pm 0.03$ | $0.77 \pm 0.05$ | – | – | – | – |
| Nature | Real Ref | $0.57 \pm 0.00$ | $0.82 \pm 0.00$ | $0.45 \pm 0.03$ | $1.19 \pm 0.06$ | $\mathbf{0.37 \pm 0.04}$ | $\mathbf{1.03 \pm 0.06}$ |
| | 75-25 | $0.56 \pm 0.01$ | $0.81 \pm 0.01$ | $0.48 \pm 0.03$ | $1.10 \pm 0.03$ | $0.39 \pm 0.03$ | $1.09 \pm 0.05$ |
| | 50-50 | $\mathbf{0.54 \pm 0.01}$ | $\mathbf{0.79 \pm 0.02}$ | $\mathbf{0.43 \pm 0.03}$ | $\mathbf{1.07 \pm 0.04}$ | $0.38 \pm 0.03$ | $1.06 \pm 0.03$ |
| | 25-75 | $0.55 \pm 0.00$ | $0.81 \pm 0.00$ | $0.47 \pm 0.03$ | $1.10 \pm 0.03$ | $0.40 \pm 0.01$ | $1.04 \pm 0.03$ |
| | Mixed11 | $0.59 \pm 0.00$ | $0.84 \pm 0.01$ | $0.54 \pm 0.06$ | $1.36 \pm 0.11$ | $0.48 \pm 0.02$ | $1.38 \pm 0.10$ |
| Sales | Real Ref | $\mathbf{0.46 \pm 0.00}$ | $0.74 \pm 0.00$ | – | – | – | – |
| | 75-25 | $0.47 \pm 0.00$ | $0.74 \pm 0.01$ | – | – | – | – |
| | 50-50 | $0.47 \pm 0.00$ | $\mathbf{0.74 \pm 0.01}$ | – | – | – | – |
| | 25-75 | $0.47 \pm 0.01$ | $0.75 \pm 0.01$ | – | – | – | – |
| | Mixed11 | $0.49 \pm 0.00$ | $0.75 \pm 0.01$ | – | – | – | – |
| Transport | Real Ref | $\mathbf{0.64 \pm 0.00}$ | $\mathbf{0.75 \pm 0.01}$ | $\mathbf{0.66 \pm 0.01}$ | $\mathbf{0.81 \pm 0.02}$ | $\mathbf{0.61 \pm 0.01}$ | $\mathbf{0.80 \pm 0.01}$ |
| | 75-25 | $0.69 \pm 0.01$ | $0.79 \pm 0.01$ | $0.91 \pm 0.01$ | $1.08 \pm 0.00$ | $0.77 \pm 0.01$ | $1.00 \pm 0.00$ |
| | 50-50 | $0.67 \pm 0.01$ | $0.78 \pm 0.00$ | $0.88 \pm 0.02$ | $1.05 \pm 0.02$ | $0.78 \pm 0.03$ | $1.00 \pm 0.03$ |
| | 25-75 | $0.67 \pm 0.01$ | $0.78 \pm 0.01$ | $0.89 \pm 0.03$ | $1.06 \pm 0.03$ | $0.76 \pm 0.03$ | $0.98 \pm 0.03$ |
| | Mixed11 | $0.74 \pm 0.00$ | $0.85 \pm 0.01$ | $1.12 \pm 0.05$ | $1.31 \pm 0.03$ | $0.95 \pm 0.03$ | $1.21 \pm 0.02$ |
| Web/CloudOps | Real Ref | $\mathbf{0.58 \pm 0.00}$ | $0.93 \pm 0.00$ | $1.05 \pm 0.09$ | $1.40 \pm 0.09$ | $1.13 \pm 0.04$ | $1.41 \pm 0.06$ |
| | 75-25 | $0.61 \pm 0.01$ | $0.91 \pm 0.02$ | $1.03 \pm 0.08$ | $1.35 \pm 0.08$ | $1.14 \pm 0.08$ | $1.37 \pm 0.05$ |
| | 50-50 | $0.60 \pm 0.03$ | $\mathbf{0.89 \pm 0.01}$ | $\mathbf{0.96 \pm 0.07}$ | $\mathbf{1.27 \pm 0.04}$ | $\mathbf{1.12 \pm 0.03}$ | $\mathbf{1.34 \pm 0.02}$ |
| | 25-75 | $0.61 \pm 0.02$ | $0.90 \pm 0.03$ | $1.02 \pm 0.13$ | $1.32 \pm 0.07$ | $1.12 \pm 0.00$ | $1.36 \pm 0.03$ |
| | Mixed11 | $0.65 \pm 0.02$ | $0.93 \pm 0.03$ | $1.09 \pm 0.13$ | $1.41 \pm 0.10$ | $1.25 \pm 0.12$ | $1.47 \pm 0.06$ |

# I. Supplementary Visualizations

## I.1. Per-Generator PCA Projections

PCA is fit on the real reference corpus only and then applied to each synthetic generator using a fixed preprocessing pipeline: we compute shared window-level features covering scale, distributional shape, autocorrelation, memory, seasonality, trend, volatility, tail/outlier frequency, changepoint density, missingness, and multivariate dependence where applicable; remove non-finite or constant dimensions; impute missing values with real-reference feature medians; standardize using real-reference scaling parameters; fit PCA on the processed real-reference matrix; and project each synthetic generator into the resulting coordinate system.

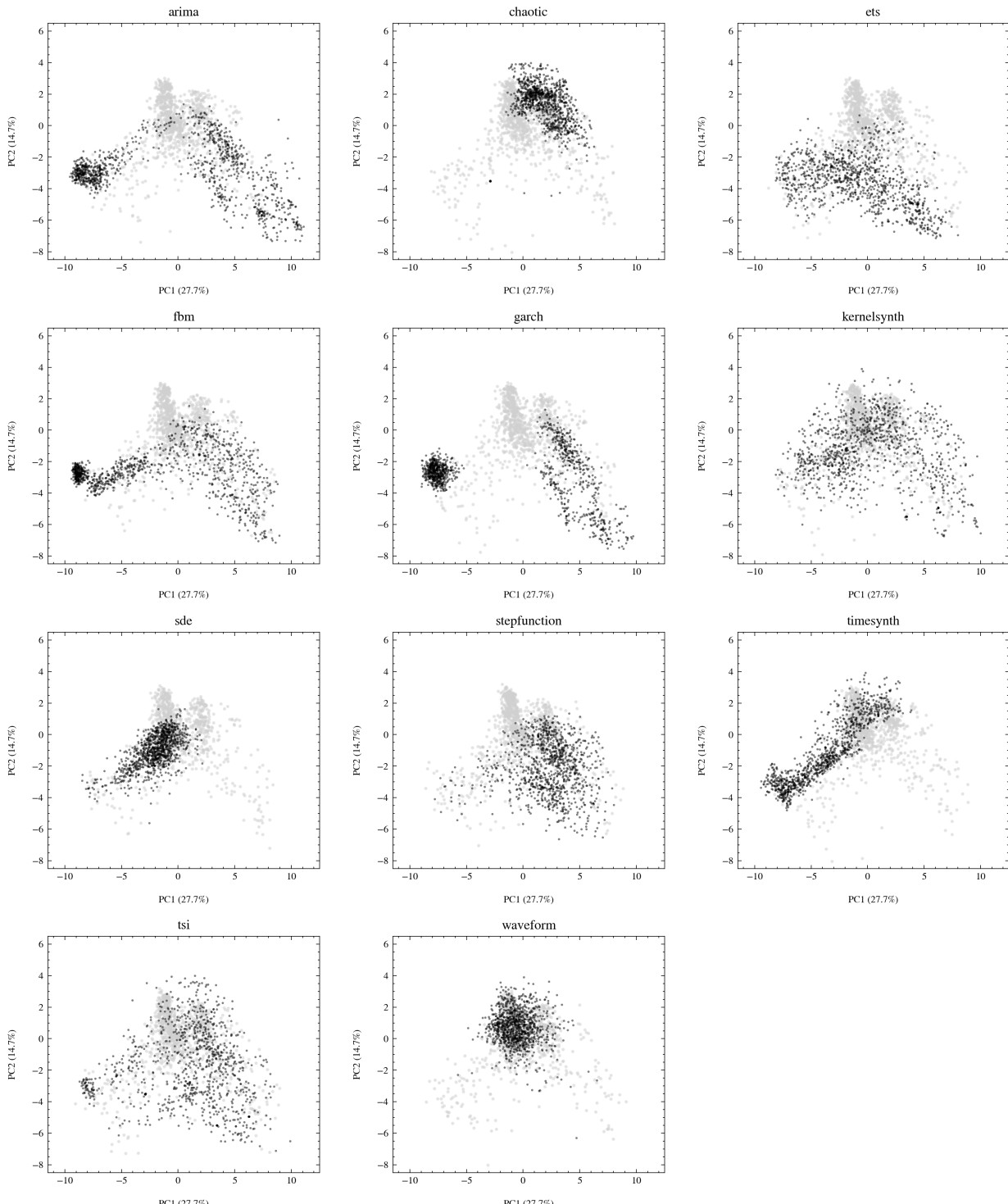

*Figure 1.* Per-generator PCA projections in the audit feature space. Each panel overlays one synthetic generator with the real reference corpus using the same PCA coordinate system. The figure is a qualitative diagnostic showing that generators occupy different regions of the audit feature space.

