# OpenReview forum: "Mix, Don’t Pick: Why Synthetic Corpus Composition Matters for Time Series Foundation Model Pretraining"
_ICML.cc/2026/Workshop/FMSD — FMSD @ ICML 2026 Poster_

### Official Review · Reviewer_GcYM · 2026-05-12
**Review for "Mix, Don’t Pick: Why Synthetic Corpus Composition Matters for Time Series Foundation Model Pretraining"**

**Rating:** 6
**Confidence:** 5

**Review:**

## Summary

This paper studies the role of corpus composition in synthetic data pretraining for TSFMs. Instead of treating synthetic pretraining as a problem of selecting one best generator, the paper evaluates multiple synthetic generators and their mixtures.

## Strengths

The paper addresses an important question: data quality and corpus construction are crucial for TSFM pretraining. The focus on synthetic data is relevant, especially given the scale, privacy constraints of real-world time-series data.

Another strength is that the paper collects and compares a relatively diverse set of synthetic time series generation methods. This provides a useful empirical reference for understanding how different synthetic sources affect downstream forecasting performance.

## Areas for Improvement

**The main concern is that the reported MASE results are quite weak**. In many settings, the MASE values are not below 1, which means the models do not clearly outperform the seasonal naive baseline. This substantially limits the practical value of the proposed pretraining corpora.
In particular, the results seem less competitive than prior evidence from CauKer [1], where Chronos-Small trained with KernelSynth synthetic data can reportedly achieve comparable performance of Chronos-Small Official. Given this comparison, the paper should better explain why the reported MASE numbers are worse, whether this comes from implementation details.

[1] Xie, Shifeng et al. “CauKer: Classification Time Series Foundation Models Can Be Pretrained on Synthetic Data.” (2025).

---

### Official Review · Reviewer_1Aat · 2026-05-19

**Rating:** 6
**Confidence:** 3

**Review:**

## Summary
The paper studies how synthetic data generator choice affects the forecasting performance of time-series foundation models. It evaluates 11 synthetic generators, real data, and real–synthetic mixtures using Chronos-T5-Mini and Moirai-Small. The main findings are that generator choice matters, rankings differ across architectures, uniform generator mixtures are strong, and the best real–synthetic mix is architecture-dependent.

## Strengths
The topic is highly relevant. Recent work shows that TSFMs trained largely on synthetic data can perform well, but there is little systematic understanding of how such synthetic corpora should be constructed. Since many existing models rely on proprietary or ad hoc recipes, this paper is a welcome step toward more systematic analysis.

The paper presents its main findings clearly. I also appreciated the use of confidence intervals, which helps communicate both the magnitude and uncertainty of the reported improvements.

## Areas for improvement
My main concern is that the study uses architectures and training setups that seem somewhat far from current state-of-the-art practice, which makes it unclear how well the conclusions transfer to newer models.

Relatedly, the absolute GIFT-Eval performance appears quite poor in many settings. For example, Chronos-T5-Mini has MASE above 1 in most experiments, meaning it is worse than the seasonal naive baseline. This is concerning given that the official Chronos-T5-Small GIFT-Eval score is 0.892, and suggests there may be an issue with the training setup, evaluation setup, or model configuration. If the models are substantially underperforming expected baselines, the conclusions about generator relevance and corpus composition become harder to interpret.

The paper would benefit from discussing this discrepancy and clarifying whether it is due to model size, training budget, implementation details, or another factor.

## Justification
I am on the edge. The topic is very relevant, the paper addresses an important gap, and the findings would likely be interesting for workshop discussion. However, the empirical performance issues raise concerns about how general the conclusions are. This would not be sufficient for a full conference paper, but I cautiously recommend acceptance for the workshop.